# High-performance 2D electronic devices enabled by strong and tough two-dimensional polymer with ultra-low dielectric constant

Qiyi Fang[1,2,8], Kongyang Yi [3,8], Tianshu Zhai[1,8], Shisong Luo[4], Chen-yang Lin[1], Qing Ai [1], Yifan Zhu [1], Boyu Zhang [1], Gustavo A. Alvarez[5], Yanjie Shao [6], Haolei Zhou[2], Guanhui Gao[1], Yifeng Liu[1], Rui Xu [1], Xiang Zhang [1], Yuzhe Wang[2], Xiaoyin Tian[1], Honghu Zhang [7], Yimo Han [1], Hanyu Zhu [1], Yuji Zhao [4], Zhiting Tian [5], Yu Zhong[2], Zheng Liu [3] ✉ & Jun Lou [1] ✉

As the feature size of microelectronic circuits is scaling down to nanometer order, the increasing interconnect crosstalk, resistance-capacitance (RC) delay and power consumption can limit the chip performance and reliability. To address these challenges, new low-$k$ dielectric ($k < 2$) materials need to be developed to replace current silicon dioxide ($k = 3.9$) or SiCOH, etc. However, existing low-$k$ dielectric materials, such as organosilicate glass or polymeric dielectrics, suffer from poor thermal and mechanical properties. Two-dimensional polymers (2DPs) are considered promising low-$k$ dielectric materials because of their good thermal and mechanical properties, high porosity and designability. Here, we report a chemical-vapor-deposition (CVD) method for growing fluoride rich 2DP-F films on arbitrary substrates. We show that the grown 2DP-F thin films exhibit ultra-low dielectric constant (in plane $k = 1.85$ and out-of-plane $k = 1.82$) and remarkable mechanical properties (Young's modulus > 15 GPa). We also demonstrated the improved performance of monolayer $MoS_2$ field-effect-transistors when utilizing 2DP-F thin films as dielectric substrates.

The improvement in the performance of modern microelectronic integrated circuits has been primarily driven by the reduction of transistor size and the increase of transistor density in a single chip[1–3]. The size feature of the transistors in a single chip rapidly scaled down to nanometer size following Moore's law in the past few decades. However, with the continuous scaling-down of transistor size, the influence of the interconnects on device performance is becoming increasingly pronounced at this nanoscale limit. The downsizing of the corresponding interconnects results in increased interconnect resistance ($R$) and capacitance ($C$) delays, leading to bigger crosstalk noise, delay in signal propagation speed and more power consumption[4]. As a result, the integration of low-dielectric-constant interlayer has become

[1]Department of Materials Science and NanoEngineering and the Rice Advanced Materials Institute, Rice University, Houston, TX 77005, USA. [2]Department of Materials Science and Engineering, Cornell University, Ithaca, NY 14853, USA. [3]School of Materials Science and Engineering, Nanyang Technological University, Singapore 639798, Singapore. [4]Department of Electrical and Computer Engineering, Rice University, Houston, TX 77005, USA. [5]Sibley School of Mechanical and Aerospace Engineering, Cornell University, Ithaca, NY 14853, USA. [6]Department of Electrical Engineering and Computer Science, Massachusetts Institute of Technology, Cambridge, MA 02139, USA. [7]National Synchrotron Light Source II, Brookhaven National Laboratory, Upton, NY 11973, USA. [8]These authors contributed equally: Qiyi Fang, Kongyang Yi, Tianshu Zhai. ✉e-mail: z.liu@ntu.edu.sg; jlou@rice.edu

crucial to reduce the RC delay and meet the increasing demand for high-frequency and high-speed signal transmission in the 90 nm node[5-7]. In order to solve the problem, the industry introduced the first carbon-doped low-$k$ dielectrics, featuring a dielectric constant ($k$-value) of approximately 3. However, its relatively high dielectric constant did not align with the requirements of ultra-large-scale integrated circuits (ICs). Subsequently, other low-$k$ dielectric materials like low-$k$ SiCOH (with $k$-values ranging from 2.5 to 2.7) were developed and gained prominence[5,8-11], but they are still far below the latest requirement of dielectric materials with ultra-low dielectric constants below 2 in the most advanced devices[2]. More importantly, most ultra-low-$k$ dielectrics exhibited inadequate mechanical performances owing to their disordered or porous structures in order to achieve the low dielectric constants[7,12,13]. However, the lack of robust mechanical properties of these ultra-low-$k$ dielectrics could lead to undesirable damages and cracking during thermal, mechanical and chemical processes involved in fabrication procedures, resulting in lower device yields, increased costs, and a compromised reliability of device performances. This strongly suggests the great urgency for developing a new ultra-low-$k$ dielectric material with robust mechanical properties.

Two-dimensional polymers (2DPs) represent a class of highly porous, covalently-linked layered polymer sheets with good mechanical and thermal properties, which is a promising candidate to meet the aforementioned requirements[14-17]. Monolayer 2DPs demonstrated remarkable mechanical properties with Young's modulus of 50 GPa and breaking strength of 6 GPa, respectively[18]. Additionally, owing to their low density and in-plane covalently-linked porous structures, 2DPs have the potential to become low-$k$ dielectric materials. It is suggested that a dielectric constant as low as 1.6 and a favorable thermal conductivity as high as $1\,W\,m^{-1}\,K^{-1}$[19], which are highly desired in high-density interconnect applications, can be achieved in 2DPs. However, to date most preparation methods of 2DP thin films are based on liquid phase synthesis, often involving corrosive solvents,

which hinder their integration into microchip fabrication processes[20-22]. On the other hand, chemical-vapor-deposition (CVD) has been widely utilized to uniformly deposit various thin film materials in microelectronic fabrication process[23-25]. Although attempts have been made to deposit 2DPs using vapor phase method, these investigations still involved corrosive organic solvent or acids, or non-uniform deposition[26-28]. Solvent-free CVD methods for the growth of 2DPs for dielectric applications remain largely unexplored.

In this work, we have successfully fabricated large-area fluoride-rich 2DP (2DP-F) film using a CVD method and conducted comprehensive measurements of its dielectric and mechanical properties. Our findings indicate that 2DP-F exhibited an ultra-low dielectric constant in both in-plane and out-of-plane direction, and its in-plane covalent bonds contribute to its superior mechanical properties, which are among the best reported for state-of-the-art low-$k$ dielectrics. The advantages of the CVD-grown 2DP-F thin films were also shown as the dielectric substrate to improve the performance of $MoS_2$ based field-effect transistor (FET).

## Results

### Synthesis and characterization of 2DP-F thin films

2DP-F films were prepared using a low-temperature CVD growth method in a double-zoned tube furnace equipped with an external heating belt, as shown in Fig. 1a. Notably, all monomers can be sublimed at relatively low temperature (Supplementary Fig. 1). Substrates together with triformylphloroglucinol (Tp) were placed downstream in a test tube and heated to different temperatures. Simultaneously, 4,4'-(Hexafluoroisopropylidene)dianiline (HFDA) from upstream was heated to 180 °C. The growth process was maintained for desired duration with a continuous flow of 100 sccm of Ar, allowing the formation of a uniform 2DP-F thin film, as shown in Fig. 1b. Atomic-force-microscopy (AFM) reveals that the CVD-grown 2DP-F film is as thin as 2 nm, with remarkable uniformity and

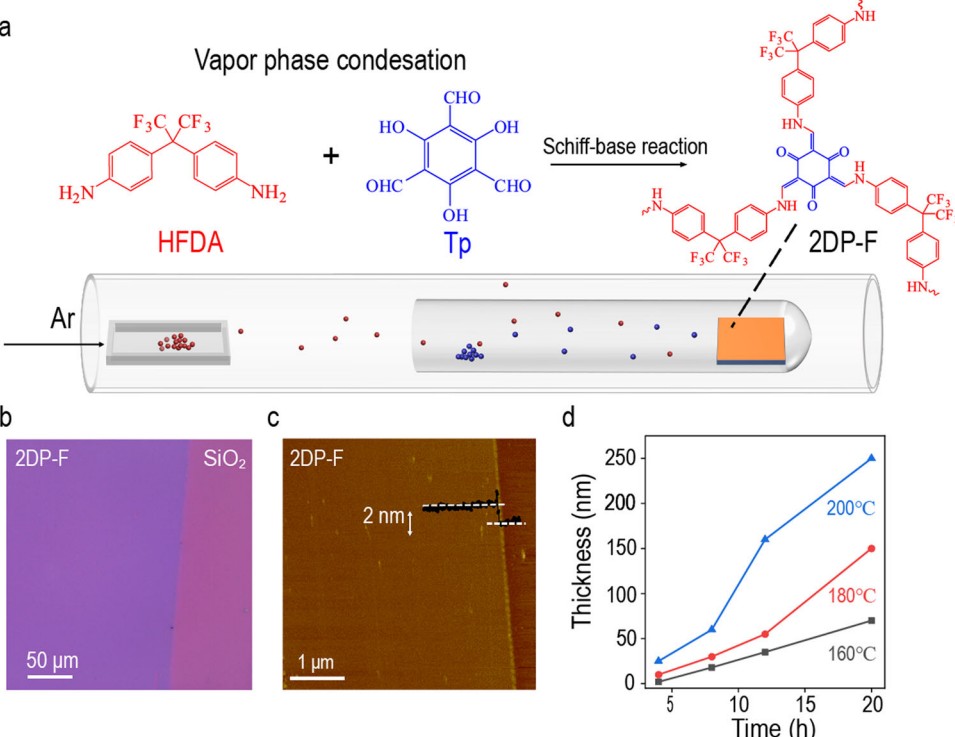

**Fig. 1 | Synthesis, structure and morphology characterization of 2DP-F film. a** Schematic illustration of the chemical-vapor-deposition (CVD) setup. **b** Optical image of a large area, uniform fluorine rich two-dimensional polymer (2DP-F) thin film. **c** Tapping mode atomic-force-microscope (AFM) image of a 2 nm thick 2DP-F film (inset: represented height profile). **d** The thickness of the CVD-grown 2DP-F thin films as a function of growth temperature and time.

minimal surface roughness (Fig. 1c). In contrast to the 2DP-F thin films grown by solution method, which exhibited rough surface and numerical precipitation particles (Supplementary Fig. 2), the continuous and stable delivery of monomers to the growth zone by the CVD method contributed to the uniform deposition of the film[29]. By varying the growth temperature and duration, the thickness of the 2DP-F films can be controlled, ranging from a few to a few-hundred nanometers (Fig. 1d), while maintaining high uniformity and smoothness (Supplementary Fig. 3). The growth rate can be increased when using 3 M AcOH in a bubbler as the catalyst and using a 5nm- thick 2DP-F film as a growth substrate. Under these conditions, the thickness of the film can reach more than 500 nm within 8 hours while maintaining a smooth surface (Supplementary Fig. 4). It is worth noting here that this method could be applied to various substrates, including SiO$_2$, sapphire, mica, quartz, gold and even plastics (polyether or polyimide), highlighting its versatility (Supplementary Fig. 5). The CVD-grown 2DP-F exhibited good thermal, mechanical, and chemical stability, and they can be patterned by standard photolithography or deposited on pre-patterned substrates (Supplementary Fig. 6).

The formation of imine bonds in 2DP-F thin film was confirmed using Raman spectra where the present of the stretching vibration of C = C groups at around 1610 cm$^{-1}$ was observed (Supplementary Fig. 7a)[30]. The X-ray photoelectron spectroscopy (XPS) results of the 2DP-F thin film show the presence of carbon, nitrogen, oxygen, and fluoride. The C1s XPS spectrum can be divided into several peaks centered at 284.4, 285.9, 288.1 and 292.3 eV, corresponding to the C-C, C-N, C-O and C-F bonds, respectively (Supplementary Fig. 7b)[31,32]. It is worth noting that the crosslink density of 2DP-F decreases as the film thickness increases. Some defects corresponding to unreacted amino groups can be observed in the N1s XPS spectrum. In a 500 nm film, the ratio of unreacted amino groups to amide groups is approximately 1:120, indicating a crosslink density of over 99.5% (Supplementary Fig. 8). The 2DP-F thin film can be easily transferred onto a transmission electron microscopy (TEM) sample grid and suspended on 50 × 50 μm holes, revealing its mechanical robustness (Supplementary Fig. 9a). The high-resolution TEM imaging and diffraction results in Supplementary Fig. 9b−d reveal that the 2DP-F thin film lacks long-range order and exhibits an amorphous structure. The 2D feature of the 2DP-F film was further confirmed by using the grazing-incident wide-angle X-ray scattering (GIWAXS) technique. A diffuse arc in the $q_z$ direction was observed, and this peak at around 1.6 Å$^{-1}$ corresponded to a layer distance of 0.393 nm, which is larger than typical interlayer spacing in 2D polymers (Supplementary Fig. 10a−c). The enlarged interlayer spacing can be attributed to the flexibility of the $sp^3$ carbon in HFDA. These results suggested that 2DP-F aligned parallel to the substrate surface. A weaker peak at around 1.0 Å$^{-1}$ was also observed, which may be attributed to the spacing between flexible chains. The amorphous structure was also confirmed by X-ray Diffraction (XRD) results (Supplementary Fig. 10d). This amorphous characteristic contributed to the flexibility of the $sp^3$ carbon in HFDA, which will reduce the out-of-plane structural rigidity[33]. The porosity nature of 2DP-F was confirmed by Brunauer-Emmett-Teller (BET) analysis. The surface area of 2DP-F powder reached 958 m$^2$/g and a pore volume of 0.57cc/g, indicating its high porosity (Supplementary Fig. 11).

## Dielectric properties of 2DP-F film

From the perspective of the molecular design in 2DP-F thin films, incorporating CF$_3$ groups is expected to decrease the dielectric constant, as widely reported in various polymeric dielectrics[34–36]. In 2DP-F thin films, the hydrophobic CF$_3$ group will effectively lower the moisture content. These flexible pedant groups hinder efficient chain packing, thus leading to amorphous structures in thin films and enhancing free volume. To evaluate the capacitance-voltage (C-V)

characteristic of 2DP-F thin films, we fabricated parallel plate capacitors on 2DP-F thin films with three different thicknesses, as illustrated in Fig. 2a (top). To prevent the short-circuit of the metal-insulator-metal (MIM) devices when using direct metal deposition, we used a "dry-transfer" method to fabricate the MIM structure by transferring Au top electrodes (TE) onto 2DP-F film coated bottom electrodes (BE) (Supplementary Fig. 12a, b)[37]. The as-fabricated device is illustrated in Fig. 2a (bottom). I-V measurement shows that 2DP-F capacitors have low leakage current under an electric field of 1 MV/cm, suggesting robust insulation properties of 2DP-F films (Supplementary Fig. 13). Figure 2b shows the C-V characteristic of 2DP-F thin films with different thicknesses at 1 MHz frequency. The dielectric constant of the 2DP-F films can be extracted from the following formula:

$$C = \frac{\varepsilon_0 \varepsilon A}{d} \quad (1)$$

Given the known electrode area A and 2DP-F thin film thickness d, the dielectric constants of 2DP-F thin films are extracted as 1.83 ± 0.10, 1.84 ± 0.16, and 1.84 ± 0.18 for 15, 21, and 33 nm film thicknesses at 1 MHz frequency, respectively. Supplementary Fig. 14 shows that for all 2DP-F thin films, the measured capacitances remain nearly constant with the voltage and slightly decrease with the frequency. The dielectric constant-frequency characteristic is shown in Fig. 2c. Taking 15 nm 2DP-F films as an example, the dielectric constant will decrease with the frequency, from 1.89 ± 0.08 at 10 kHz to 1.83 ± 0.10 at 1 MHz (Fig. 2c). We also fabricated a MIM structure on a thicker film (130 nm) using the direct metal deposition method. The 130 nm 2DP-F film exhibited a dielectric constant of 1.92 and a breakdown field of 1.1 MV/cm (Supplementary Fig. 15a−c). The increased dielectric constant and decreased breakdown field might be attributed to the unreacted amino groups or hydroxyl groups (Supplementary Fig. 8)[38].

The parallel plate capacitor measurement indicates that 2DP-F thin film can be used as low-$k$ dielectric material in the out-of-plane direction. To highlight the potential of 2DP-F as an interconnect dielectric (ICD) layer, we also measured the in-plane dielectric constant by depositing 2DP-F thin films in interdigitated capacitors on sapphire substrates (Fig. 2d)[39]. The gap distance is 100 nm, as indicated by scanning electron microscopy (SEM) (Supplementary Fig. 16). After deposition, the 2DP-F can be easily filled into the gaps in the capacitors by the CVD method, which is confirmed by the bright-field TEM image and EDS elemental mapping (Fig. 2e) of the cross-section of the capacitors. The in-plane dielectric constant can be extracted by comparing capacitance before and after the gap filling (Supplementary Fig. 17a) and the by Silvaco TCAD simulation[40]. Figure 2f shows the dielectric constant-capacitance relationship in our given device morphology at 1 MHz. The 2DP-F thin film shows an average in-plane dielectric constant of 1.85 ± 0.16; the in-plane breakdown voltage is 300 kV/cm (Supplementary Fig. 17b).

To evaluate the impact of using 2DP-F as ICD to reduce the RC delay constant, a Silvaco TCAD simulation is carried out[40]. The simulated structure is shown in Supplementary Fig. 18a. In this simulation, the length of the metal wires is 100 nm, and the width and height of the metal wires are fixed at 8 nm. The total parasitic capacitance in the middle metal wire is simulated when applying different ICDs with different dielectric constants.

As demonstrated in Supplementary Fig. 18, the total capacitance is reduced by 52.8% and 28.9% when replacing silicon oxide (k ~ 4.0) and SILK® (trademark of The Dow Chemical Company, $k$ ~ 2.6) with 2DP-F ($k$ ~ 1.8) as the ICD, respectively, as the device feature scale down (Supplementary Fig. 18b). The resistance of the simulated aluminum wire is 44 Ω, resulting in a reduction of the RC delay constant ($\tau = R \times C$) from 8.67 × 10$^{-16}$ s and 5.76 × 10$^{-16}$ s to 4.09 × 10$^{-16}$ s when replacing the interconnect silicon oxide and SILK® with 2DP-F, respectively. These

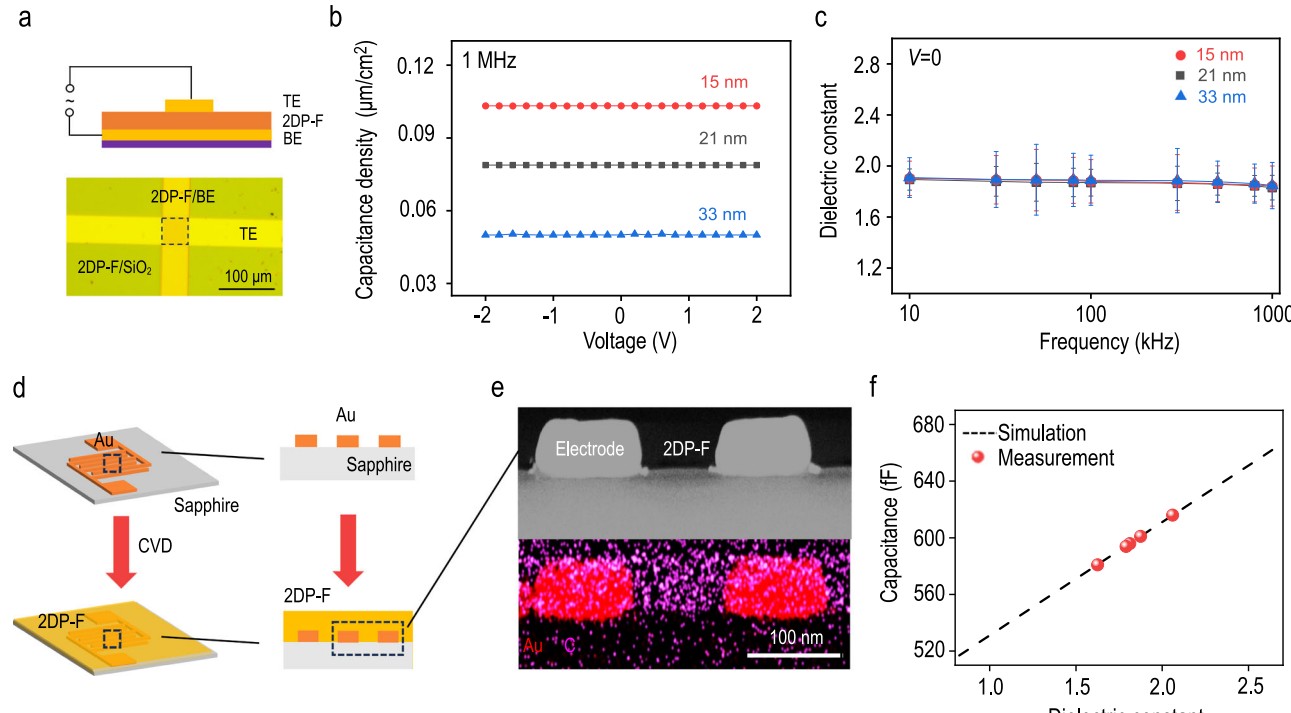

**Fig. 2 | Dielectric properties of 2DP-F films. a** Schematic illustration of the structure (top) and optical image (bottom) of the parallel plate capacitor. 2DP-F film was directly grown on bottom electrode (BE) and top electrodes (TE) were transferred on top of the film. **b** Voltage-dependent capacitance ($C–V$) and (**c**) Relative dielectric constant as a function of frequency ($C–f$) for parallel plate capacitors of 2DP-F thin films with varied thicknesses. The error bars were based on five devices for each thickness. **d** schematic illustration of 2DP-F thin films grown in the gaps of interdigitated capacitors. The Right figure illustrates the cross-section of interdigitated capacitors in the dashed square. **e** Transmition electron microscope (TEM) image (top) and the energy dispersive X-ray spectroscopy (EDS) elemental mapping of the cross-section of the interdigitated capacitors. **f** Dielectric constant determination from the Silvaco TCAD simulation. The black dashed line corresponds to the dielectric constant of the interdigitated capacitors as a function of the dielectric constant of the gap-filling materials. Red dots correspond to the measured capacitance of the capacitors after filling the gaps with 2DP-F materials. Inset is a modeled equivalent circuit of impedance behavior in the interdigitated capacitors.

simulation results highlight the potential of 2DP-F as an interconnect dielectric to reduce the RC delay constant as the features of the devices scale down.

## Mechanical properties of 2DP-F film

Due to the presence of strong in-plane covalent bonds in 2DPs, 2DP-F is expected to exhibit superior mechanical robustness over previous low-$k$ dielectrics[17,18]. To measure the Young's modulus (YM) and breaking strength of 2DP-F by a nanoindentation method, 2DP-F thin films with varying thicknesses were transferred onto a TEM sample grid with holey carbon arrays[41]. The SEM and AFM images show the good suspension of 2DP-F thin films over the holes (Fig. 3a, b). A schematic illustration of the nanoindentation test is provided in Supplementary Fig. 19a and the insert in Fig. 3c, where the AFM tip is centered at the suspended region and pressed downward to deform the atomically-thin 2DP-F samples until they break. The load versus indentation depth curves are recorded as shown in Fig. 3c. After the indentation, we noticed that the indentation rupture was highly localized as shown in Fig. 3d, indicating the non-crystalline structure of 2DP-F[42]. The load-displacement curves can be obtained by subtracting the cantilever deflection (Supplementary Fig. 19b), and the 2D modulus and 2D stress can be extracted by fitting the load-displacement curves. The distribution of 2D modulus $E^{2D}$ and 2D stress $\sigma^{2D}$ for 2DP-F thin films are shown in Fig. 3e, and the corresponding Young's modulus and breaking strength can be obtained by dividing $E^{2D}$ and $\sigma^{2D}$ by the thickness of the film (Fig. 3f). For an 11 nm 2DP-F thin film, the Young's modulus and breaking strength are $16.8 \pm 2.9$ and $1.01 \pm 0.08$ GPa, respectively. As the film thickness increases, the Young's modulus and breaking strength of 2DP-F gradually decrease. The 30 nm 2DP-F thin film

exhibited Young's modulus of $15.6 \pm 1.8$ and $0.96 \pm 0.12$ GPa, respectively. The decrease in mechanical properties is possibly attributed to the weak interlayer interactions observed in many 2D materials and the increased defects as the thickness increases (Supplementary Fig. 8)[43]. We also measured a 50 nm 2DP-F thin film by tensile testing method using a nanomechanical device, resulting in a Young's modulus of 12.7 GPa and breaking strength of 0.8 GPa, respectively, which is consistent with the trend observed here (Supplementary Fig. 20).

In the realm of low-$k$ dielectrics, a fundamental trade-off exists between achieving a low dielectric constant and maintaining good mechanical stability. The high porosity and low-density in low-$k$ dielectrics typically lead to significant deterioration of the mechanical strength. For instance, Young's modulus of $SiO_2$ declines sharply from 76 GPa to below 10 GPa as porosity increases to 50%[7]. As a porous polymer, 2DPs have been proven to exhibit superior mechanical properties despite their low density and high porosity. To highlight the superior mechanical performance of 2DP-F, we made a comparison with other low-$k$ dielectric thin films with 2DP-F[39,44–53]. Figure 3g illustrates that 2DP-F falls into the ultra-low-$k$ category ($k < 2.0$), displaying superior normalized Young's modulus when compared to other low-$k$ dielectric films, particularly linear polymers with dielectric constants exceeding 2 and Young's moduli below 5 GPa. This again is attributed to the high degree of covalent cross-linking and two-dimensional nature of 2DP-F. While porous silicon zeolites (PSZs) occupy the attractive top left corner in this context, the brittle nature of PSZs as the crack will significantly reduce the breakdown voltage of the dielectrics, and the scalable production of uniform PSZ thin films at nanoscale dimensions remains unexplored. In contrast, 2DP-F film is more fracture tolerant[17], and the preparation of 2DP-F thin films

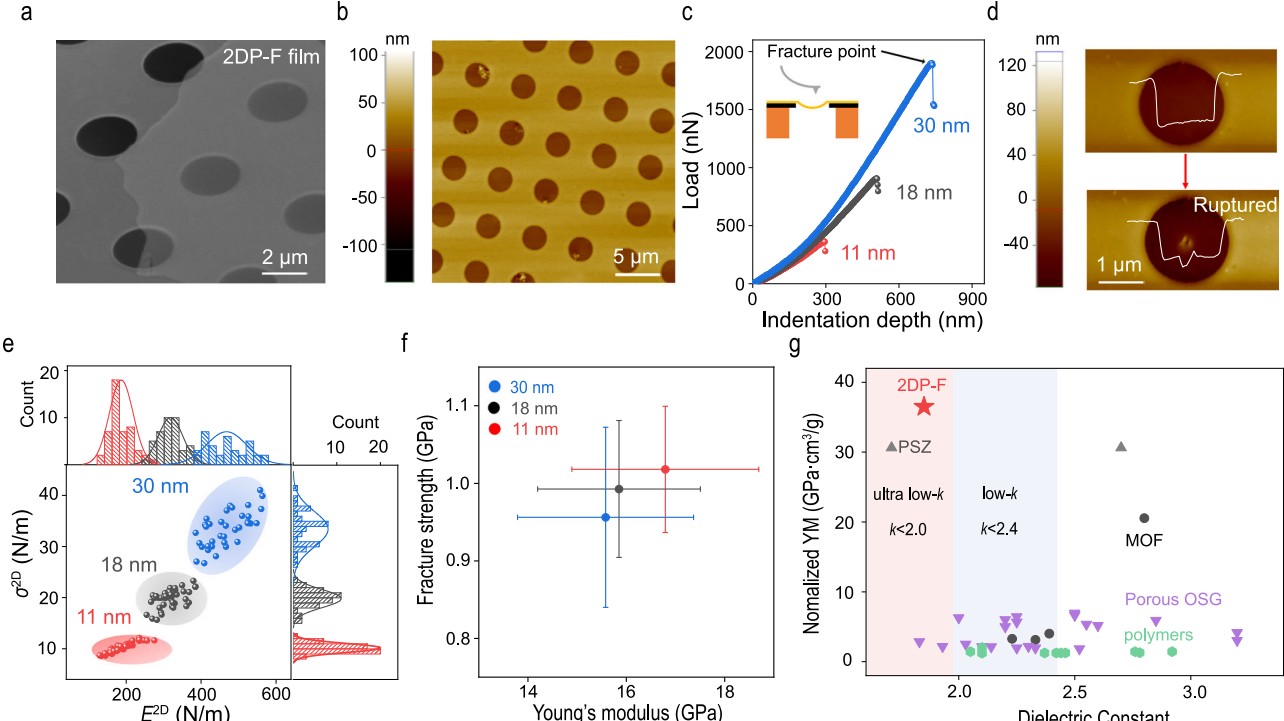

**Fig. 3 | Mechanical properties of 2DP-F films. a** Scanning electron microscope (SEM) image of 2DP-F thin film suspended over a holey substrate. **b** Tapping mode AFM image of the suspended 2DP-F thin film. **c** Load versus indentation depth curves of the 2DP-F thin films with varied thicknesses. Inset demonstrated the AFM indentation experiment on a suspended film. **d** Tapping mode AFM image of suspended 2DP-F film and corresponding line profile before (top) and after (bottom) indentation experiment. **e** Statistic histogram of the 2D modulus and the 2D stress of 2DP-F thin films, with more than 40 data points measured for each sample condition. **f** Corresponding Young's modulus and breaking strength of 2DP-F thin films. The error bars were based on more than 30 data point for each thickness. **g** Comparisons of many low-*k* dielectric materials as mapped in the normalized Young's modulus vs. dielectric constant plot[39,44–53].

through a solvent-free, low-temperature CVD method offers a distinct advantage for its application as an ultra-low-*k* dielectric material in microelectronic devices. In addition to its superior mechanical properties, 2DP-F also exhibits good out-of-plane thermal conductivity of 0.38 W/m·K (Supplementary Fig. 21). This value is comparable to many metal organic framework (MOF) materials and higher than most low-*k* polymers[54–65]. However, due to the increased interlayer distance with the introduction of the flexible HFDA building blocks in 2DP-F, this value is lower than that of crystalline 2D COFs with better long-range order and preferred orientation[19].

## MoS₂ FET performances on 2DP-F substrates

In addition to the ultra-low dielectric constant nature of 2DP-F, the dangling bond free and uniform surface of CVD grown 2DP-F has provided the possibility of using them in large-area modification of different substrates for device fabrications, which has been widely recoginized in 2D h-BN devices. Drastic improvements were observed in the CVD grown MoS₂ based FET performances using 2DP-F film as dielectric substrate. To investigate the electrical properties of the monolayer MoS₂/2DP-F thin film heterostructures, back-gated FETs were fabricated by transferring CVD-grown monolayer MoS₂ crystals (Supplementary Fig. 22) onto the 2DP-F thin film directly grown on HfO₂/ n++ Si substrate, followed by photolithography and electrodes deposition. Figure 4a shows the optical image of the fabricated device (bottom) and its configuration schematics (top). The *I-V* characteristics of a typical device with a channel length of 30 μm and channel width of 4 μm, measured under ambient conditions, is shown in Fig. 4b. The linear feature of the output curves near $V_{ds} = 0$ suggests the formation of ohmic contact between MoS₂ and the source and the drain electrodes. The transfer curves of the MoS₂ FET exhibited n-type conductance with an average mobility of $10.4 \pm 1.8$ cm²/V·s, (Fig. 4c) while the largest on/off ratio was estimated to be ~ 10⁷. As a comparison,

MoS₂ directly transferred on HfO₂/Si substrates exhibited a mobility of $4.5 \pm 1.2$ cm²/V·s and largest on/off ratio of ~ 10⁶ (Fig. 4d). The 2DP-F based devices also show significantly decreased hysteresis (Supplementary Fig. 23), indicating fewer charge traps at the interface. Using Such an impressive increase in the device performance was mainly attributed to the decrease in the potential charge scattering by dangling bonds or charge traps from the oxide substrate, which was widely recognized in oxide dielectrics for 2D FETs[66,67]. To further probe the interface properties, the trap density ($D_{it}$) between MoS₂/2DP-F and MoS₂/HfO₂ was estimated using the following expression from the subthreshold swing (SS):

$$SS = \frac{\ln(10)k_bT}{q}\left(1 + \frac{qD_{it}}{C_G}\right) \quad (2)$$

Where $k_b$ is the Bolzman constant, $q$ is the elementary charge and $C_G$ is the gate capacitance. The extracted $D_{it}$ is around $6.2 \times 10^{13}$ cm⁻²eV⁻¹ and $1.1 \times 10^{14}$ cm⁻²eV⁻¹ for MoS₂/2DP-F and MoS₂/HfO₂ interface, respectively. These results indicate that the CVD-grown 2DP-F thin film is a robust dielectric material with an attractive combination of multiple functions.

In conclusion, we have successfully prepared fluoride-rich 2DP-F thin films with controllable thickness using a simple solvent-free CVD method on various substrates. The CVD-grown 2DP-F thin films exhibited an ultra-low dielectric constant of approximately 1.8. Furthermore, the solvent-free and gap-filling nature of the deposition process highlights its potential for future applications in microelectronics fabrication processes. Distinct from most low-*k* dielectrics, 2DP-F also exhibited superior mechanical properties, likely due to its two-dimensional nature and the presence of strong in-plane covalent bonds. Additionally, the dangling bond-free surface of 2DP-F thin films successfully demonstrated the potential to enhance the performances

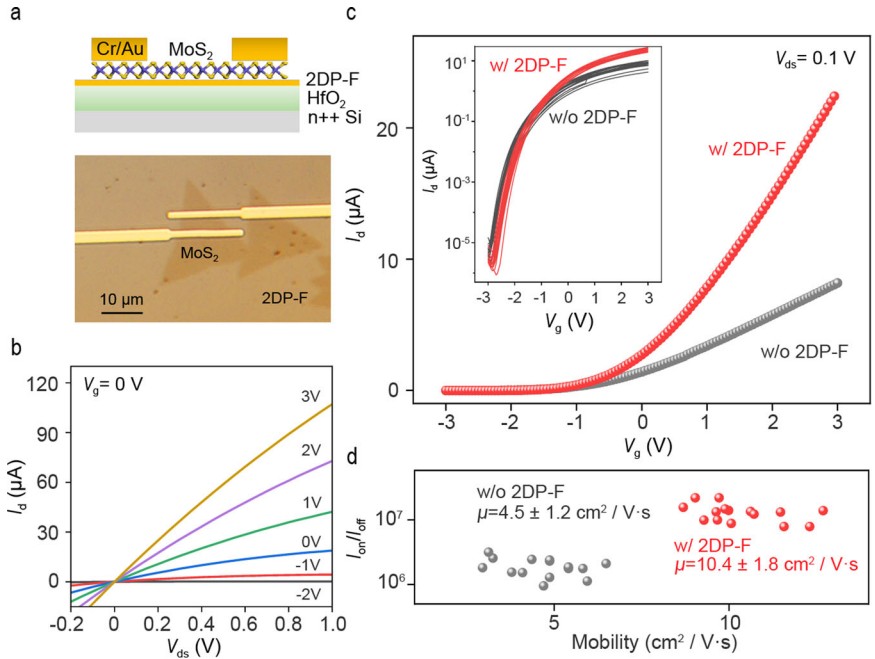

**Fig. 4 | Electrical properties of MoS$_2$ field-effect-transistors (FETs) based on 2DP-F films. a** Optical image of the FET device fabricated on transferred monolayer MoS$_2$/2DP-F thin film heterostructure. **b** The Drain current ($I_d$) - Drain voltage ($V_{ds}$) characteristic of the FET devices, gate voltage ($V_g$) = 0. **c** Transfer characteristics of the devices with and without 2DP-F dielectric layer. Inset: log-scale transfer characteristic of the devices with and without 2DP-F. **d** The distribution of the mobility and on-off ratio of the FET devices with and without the 2DP-F dielectric layer.

of MoS$_2$ FET devices by minimizing the surface charge scattering. This work represents an important step forward for the integration of 2DPs in high-performance 2D electronic devices, opening up new possibilities for practical applications of 2DPs.

## Methods

### CVD growth of 2DP-F thin films
The CVD growth of 2DP-F thin films was conducted in a two-zone electric furnace equipped with a two-inch quartz tube and a gas flow system controlled by a mass flow controller. Typically, 5 mg of HFDA monomer (Sigma-Aldrich, 98%) was added to a quartz boat and placed at the first heating zone. 5 mg of Tp (Ambeed, 95%) together with desired substrates were placed in a test tube (inner diameter around 20 mm) and the tube was placed in the second zone with the open end facing upstream. The system was first purged with 200 sccm of Ar to remove residual oxygen, followed by raising the temperature of two heating zones to the desired temperature in 30 min to start the growth. The duration of the growth varied from 4 to 20 h to obtain 2DP-F thin films with different thicknesses. After the reaction, the furnace was cooled down naturally.

### Grazing-incidence wide-angle X-ray scattering (GIWAXS)
GIWAXS experiments were carried out at the 11-BM Complex Materials Scattering (CMS) beamline of National Synchrotron Light Source II (NSLS-II), Brookhaven National Laboratory. The samples were measured in vacuum with an X-ray beam with a photon energy of E = 13.5 keV (λ = 0.9184 Å) and an incident angle of 0.1°. The scattering signal was collected by an in-vacuum 2D detector (Pilatus 800k, 172 × 172 μm$^2$ per pixel). The sample-to-detector distance was 260 mm, which was calibrated using silver behenate. The data acquisition time was 10 s.

### Transfer of 2DP-F
A PMMA-assisted transfer method was used to transfer 2DP-F thin films onto different substrates, including Au coated n + + Si, Holey carbon TEM grid etc. The as-grown 2DP-F on mica thin films was coated with

PMMA A4 by spin-coating (1000 rpm) for 2 mins, and baked at 120 °C for at least 30 mins. A thermal release tape (TRT, Nitto) was gently pressed on the PMMA/2DP-F/Mica substrate, and then immerse the TRT/PMMA/2DP-F/Mica in water. The TRT/PMMA/2DP-F will detach from the mica substrate within minutes. The TRT/PMMA/2DP-F film was then transferred onto desired substrates and heated to 60 °C to ensure good adhesion of 2DP-F thin film to the substrates. The TRT layer was removed by heating the TRT-PMMA/2DP-F films to 120 °C. And the PMMA layer was removed by soaking the substrate in acetone for 1 hour, leaving the 2DP-F film on desired substrates.

### Impedance measurements
Impedance measurements were carried out in a Cascade Summit 12 K probe station equipped with Agilent 4294 A precision impedance analyzer. 2DP-F was directly grown on the bottom eletrodes, followed by transferring a E-baem deposited gold electrode with a thickness of 60 nm by a PMMA assisted method. *C-f* measurements were performed at 0 d.c. bias, and *C-V* measurements were conducted at 10 kHz to 1 MHz.

### Fabrication of interdigitated electrodes
PMMA A4 was spin coated on sapphire substrate at 4000 rpm for 70 s and baked at 150 °C for 2 min, and then patterned by electron beam lithography method. 3/47 nm Ti/Au was deposited by e-beam evaporation and lift off by ultrasonic treatment in acetone. The interdigitated capacitor on sapphire was directly used as the growth substrate to fill the gaps with 2DP-F. The in-plane capacitance was measured using a Keysight E4980A LCR meter.

### 2D capacitance simulation
The capacitance is calculated by Silvaco TCAD. The substrate, medium and electrode bars' size, location, and materials parameters are defined to simulate the devices we measured. The boundary conditions are grounded and positive bias for two electrode bars separately. The appropriate size of the mesh is selected to divide the simulation region. Finally, we obtain the relationship between the capacitance and the medium's permittivity.

## Nanoindentation test

Single crystal diamond tips (K-Tek, D80) were used for nanoindentation test in a Park NX20 AFM system. The spring constant of each tip was calibrated by a thermal noise method. During the nanoindentation test, the indentation speed was controlled at 0.1 μm/s.

## MoS₂ FET measurement

CVD-grown monolayer $MoS_2$ was transferred onto $HfO_2$/n+ + Si substrate with or without 2DP-F thin film using the standard PMMA-assisted transfer method. After $MoS_2$ monolayer was transferred onto the substrate, electrodes were patterned by photolithography. 5/30 nm Cr/Au was deposited by e-beam evaporation and lift off by aceton. The FET performance was measured using Keysight B1500A semiconductor device parameter analyzer in a high vacuum of $\sim 10^{-4}$ Torr.

## Data availability

The data findings of this study are available from the corresponding authors on request.

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

## Acknowledgements

This work was supported by the AFRL International RISING (Research, Innovation & Science In Nanotechnology) Center at Rice University and the NSF I/UCRC Center for Atomically Thin Multifunctional Coatings (ATOMIC) under award # EEC-2113882. Z.L. and K.Y. acknowledge the support from National Research Foundation–Competitive Research Programme NRF-CRP22-2019-0007,NRF-CRP22-2019-0004 and NRF-CRP26-2021-0004, the Singapore Ministry of Education Tier 3 Pro-gramme "Geometrical Quantum Materials" AcRF Tier 3 (MOE2018-T3-1-002), and A*STAR MTC Programmatic Grant M23M2b0056. G.G. and Y.H. acknowledge NSF (CMMI–2239545) and Welch Foundation (C-2065) support. Z.T. and G.A. acknowledge the support of the Office of Naval Research grant (N00014-22-1-2357) and the NSF Graduate Research Fellowship with the grant number of 1650114. S. Luo and Y. Zhao are supported in part by ULTRA, an Energy Frontier Research Center funded by the U.S. Department of Energy (DOE), Office of Sci-ence, Basic Energy Sciences (BES), under Award No. DE-SC0021230; and in part by CHIMES, one of the seven centers in JUMP 2.0, a Semi-conductor Research Corporation (SRC) program sponsored by DARPA. This research used the Complex Materials Scattering (CMS) 11-BM beamline of the National Synchrotron Light Source II, a U.S. Department of Energy (DOE) Office of Science User Facility operated for the DOE Office of Science by Brookhaven National Laboratory under Contract No. DE-SC0012704.

## Author contributions

J.L. conceived the idea. Q.F., T.Z. and Y. Zhu, carried out the 2DP-F synthesis and the mechanical characterization experiments and ana-lyzed the results. K.Y. conducted device fabrication and electrical measurements under the supervision of Z.L. H. Zhou. conducted the MIM device fabrication. Y.S. performed capacitance measurement. T.Z., R.X., and C.L. conducted MoS$_2$ synthesis and FET fabrication. Q.A., B.Z., Y.L., Y.W., X.Z. and X.T. performed material characterizations. H. Zhang. conducted GIWAXS measurement. S. L. performed device simulation under the supervision of Y. Zhao.; G.G. and Y.H. performed TEM char-acterization. G.A. conducted FDTR measurement under the supervision of Z.T. H. Zhu and Y. Zhong provided helpful discussion. Q.F., K.Y., T.Z., and J.L. wrote the manuscript. All authors discussed the results and commented on the manuscript.

## Competing interests

The authors declare no competing interests.
