## [Transparent Peer Review file · Nature Communications]

High-performance 2D electronic devices enabled by strong and tough two-dimensional polymer with ultra-low dielectric constant

Corresponding Author: Professor Jun Lou

Version 0:

Reviewer comments:

Reviewer #1

(Remarks to the Author)

Fang et al. developed two-dimensional, fluorine-rich polymer dielectric material composed of HFDA and Tp (2DP-F), which showed ultra-low dielectric constant below 2.0. This polymer dielectric material was synthesized via chemical vapor deposition, enabling uniform fabrication compared to solution-based method. Even with the ultra-low dielectric constant, the 2DP-F also exhibited the excellent mechanical properties, which can be regarded as one of the best performances among the reported low-k polymer dielectric materials. This study proposed polymer dielectric materials and suitable process technologies for the utilization as low-k interconnects, and also provided design rules for ultralow-k materials. However, some additional experimental data or explanation should be supplemented:

1. In Fig. 1d, the thickness has linear relationship with time, but it seems that there is a ramp time. For example, the deposited thickness is negligible after 2~3 hours at the growth temperature of 160 and 180 C. Can authors explain them in terms of the deposition mechanism?
2. For low-k interconnects, the thickness higher than micrometer is typically recommended. However, the deposition rate of 2DP-F is quite slow. Is there explanation or strategy how the deposition rate can be improved?
3. The authors claimed that the 2DP-F is an amorphous structure, but there are some clear peaks in XRD data (Fig. S7). What is the origin of the peaks?
4. As the dielectric constant showed the impressive values, they should be firmly confirmed. It seems the dielectric constant values were measured by the parallel plate capacitors with Al₂O₃. I suggest that the authors should measure the dielectric constant values of 2DP-F without Al₂O₃ and confirm that it shows consistent results. The leakage current issue can be excluded in sufficiently large thickness of 2DP-F.
5. Please make sure that the I-V characteristics of capacitors in Fig. S10 were obtained with Al₂O₃ or 2DP-F only. Also, Y-axis should be represented by leakage current per unit area (leakage current density) rather than leakage current. What about the out-of-plane breakdown field of the 2DP-F?
6. The dielectric constant was measured to be slightly, but consistently increased according to the thickness of the 2DP-F. The explanation should be provided.
7. The authors claimed that the excellent mechanical properties can be attributed to the covalently crosslinked structure and 2-dimensional structure of 2DP-F. Can crosslinking density be calculated through chemical analysis? Also, according to the authors' explanation and analysis, I understood that 2DP-F has a horizontal (face-on) orientation in two dimensions. Could you analyze the orientation of 2DP-F or provide a suitable scheme and explanation?
8. The 2DP-F exhibited ultra-low-k as well as superior mechanical properties compared to polymers and porous OSGs (Fig. 3g). It has been reported that 2D hexagonal boron nitride (hBN) has low dielectric constant and excellent mechanical properties. Can the authors describe the advantages of 2DP-F compared to 2D hBN?
9. It should be better that the trap density of 2D MoS₂ FETs with 2DP-F is quantitatively analyzed compared to that without 2DP-F.
10. Compared to those without 2DP-F, the mobility variation of 2D MoS₂ FETs with 2DP-F is considerable. Based on the results, it seems that there are potential issues with the process uniformity or reproducibility of 2DP-F.

Reviewer #2

(Remarks to the Author)

Summary:

Fang and co-authors reported on the development of a vacuum-deposited two-dimensional polymer (2DP) material, which exhibit low-k dielectric value of approximately 2. Using double-zoned tube furnace and reacting 4,4'-(Hexafluoroisopropylidene)dianiline (HFDA) with triformylphloroglucinol (Tp)/substrate, the authors achieved growth of low-k 2DP film on arbitrary substrate via chemical vapor deposition (CVD). The authors demonstrated that the transistors employing the proposed dielectric exhibit superior electrical performances (i.e. on-current level, carrier mobility, and etc) as compared to those built without their dielectric. However, despite the novel findings of vacuum-deposited 2DP film, the current version of manuscript is not appropriate for publication in Nature Communications unless the authors provide better demonstration of how their method and material is beneficial in terms of device fabrication, as stated in their introductory section. Specific technical comments are as follows:

Technical Comments:

Comment #1: The authors mention in the introduction that for integrated circuits (ICs), the development of low-k dielectric with robust mechanical properties are desired to reduce the RC delay. Yet in their manuscript, there is no evidence that their developed low-k dielectric material is capable of achieving such reduced RC delay in ICs. Please provide proper evidence that the proposed material can be used to achieve such breakthrough. It is hard to find proper correlation between what the authors are claiming and what the authors are actually presenting.

Comment #2: One of the main reasons for developing vacuum-based deposition method for 2DPs is believed to be associated with compatibility to current microfabrication techniques for device fabrication, to grant large-area uniformity, sequential and in-situ deposition of other functional layers, followed by their processing via techniques such as photolithography, etching, and etc. For this, the developed dielectric requires high stability in terms of chemical, thermal, and mechanical performances. These aspects are missing in the manuscript so please provide relevant data.

Comment #3: In addition to Comment #2, the authors need to provide statistical data in terms of device uniformity (transistor performance) when produced in large-scale. It appears in Figure 4d that there is a substantial variation in the carrier mobility of transistors fabricated using the proposed dielectric, and the authors need to specify whether it is attributed to the dielectric performance or other causes are involved.

Comment #4: Once again, as briefly commented in Comment #2, one of the main reasons for developing vacuum-deposited dielectric layer should be compatibility with current microfabrication techniques. It seems from the structures and fabrication methods for the parallel plate capacitors and transistors that when metal electrodes are directly deposited onto the proposed dielectric layer, it may be severely damaged. Please clarify this issue.

Reviewer #3

(Remarks to the Author)

Fang et al report a chemical-vapor-deposition method for growing two-dimensional polymer on arbitrary substrates. While the properties of the polymers are consistently well-regulated, the manuscript still leaves some room for doubt.

1. In the manuscript, the author defines 2DP-F as a two-dimensional polymer. Whether the polymer has two-dimensional molecular structure, not a cross-linked polymer, can it be proven with experimental data? In addition, it's necessary to supplement the pore size and porosity of 2DP-F for analyzing whether there are so many micropores contributing to the low dielectric constant performance of the polymer.
2. The dielectric properties of 2DP-F are affected by the presence or absence of unreacted hydroxyl and amino groups.
3. In the chapter analyzing the mechanical properties of the 2DP-F film, as the thickness of the polymer film increases, the mechanical performance decreases. The author explains this is due to weak interlayer interactions. However, it is also possible that 2DP-F is a cross-linked polymer, and as the thickness increases, the defects increase, leading to the degradation of the entire film's mechanical strength.
4. In the introduction, the author emphasizes the potential of the high thermal conductivity of 2D polymers in high-density interconnect applications. May I know what are the in-plane and through-plane thermal conductivities of 2DP-F? Are they higher than the reported low-dielectric constant polymers?
5. In Figure S10, it can't be ruled out that the low leakage current density is contributed by the layer of aluminum oxide, rather than the body of the polymer film.
6. In this sentence "which is confirmed by the bright-field TEM image and EDS elemental mapping (Figure 3e) ", the Figure 3e does not correspond to the related description.

Version 1:

Reviewer comments:

Reviewer #1

(Remarks to the Author)

I acknowledge the authors have addressed the questions and comments that I raised, by providing additional experimental evidence and explanations. The results from the original manuscript, along with the additional data provided during the revision process, strongly support that the 2DP-F polymer developed in this study exhibits a low dielectric constant in addition to excellent electrical and mechanical properties. Furthermore, the authors have demonstrated its practical applicability by applying the 2DP-F polymer to 2D MoS₂ transistors. Therefore, I now agree to publish this revised manuscript in Nature Communications.

I was also asked to comment on the authors' revisions in response to the concerns raised by referee #3 during the previous round of review.

In the 1st round of revision for Fang et al., the Reviewer #3 evaluated that the manuscript is well-regulated, but suggested some comments to improve the manuscript, as follows.

In the 1st comment, the Reviewer suggested to provide experimental evidence of two-dimensional structure of the 2DP-F polymer and to supplement the porosity of 2DP-F with its pore size. The authors provided the GIWAXS data to prove its two-dimensional structure with the proper reference and provided BET analysis data to present its high porosity with the exact value of pore volume.

In the 2nd comment, the Reviewer mentioned that the dielectric properties of 2DP-F can be affected by unreacted hydroxyl and amino groups. The authors provided the experimental data that confirmed the amount of unreacted amino groups increased with the increasing thickness of the 2DP-F polymer. Nevertheless, most of amino groups reacted in the thickest film (~500 nm), and the dielectric constant was still very low (~1.92) with the increased thickness of 2DP-F (~130 nm).

In the 3rd comment, the Reviewer suggested that the decreased mechanical performance of the 2DP-F polymer with increased thickness may result from increased defects. The authors acknowledged this comment and included this possibility in the revised manuscript.

In the 4th comment, the Reviewer required the thermal conductivity of the 2DP-F. The authors analyzed thermal conductivity by TDTR experiment, and provided out-of-plane thermal conductance of the 2DP-F, which can be regarded as a higher value than most low-k polymers. One concern is that the references on thermal conductivity values for low-k polymers were outdated. I suggest that the authors find recent studies on low-k polymers and compare the thermal conductivity value. Also, the Reviewer was curious about in-plane and through-plane thermal conductivity values. I recommend that the authors provide the detailed explanation regarding technical limitations.

In the 5th comment, the Reviewer mentioned that the leakage current density can be affected by aluminum oxide layer. The authors made a reasonable response by providing leakage current property of the MIM device with 2DP-F polymer.

In the last comment, the Reviewer pointed out the typo on the figure number, and the authors revised it.

Reviewer #2

(Remarks to the Author)

The authors have well prepared their revised manuscript, and it appears that the concerns raised by the reviewer has been clarified with empirical and simulation data. Although it is still a bit ambiguous as to how the developed method and material is beneficial in terms of device fabrication, the design rules for ultralow-k dielectric are valid and maybe used to explore ICs with low-k interconnects. In this regard, the reviewer feels that the manuscript deserves publication in Nature Communications without further revisions.

REVIEWER COMMENTS

Reviewer #1 (Remarks to the Author):

Fang et al. developed two-dimensional, fluorine-rich polymer dielectric material composed of HFDA and Tp (2DP-F), which showed ultra-low dielectric constant below 2.0. This polymer dielectric material was synthesized via chemical vapor deposition, enabling uniform fabrication compared to solution-based method. Even with the ultra-low dielectric constant, the 2DP-F also exhibited the excellent mechanical properties, which can be regarded as one of the best performances among the reported low-k polymer dielectric materials. This study proposed polymer dielectric materials and suitable process technologies for the utilization as low-k interconnects, and also provided design rules for ultralow-k materials. However, some additional experimental data or explanation should be supplemented:

Respond: We thank the reviewer for reviewing our manuscript and recognized that "Even with the ultra-low dielectric constant, the 2DP-F also exhibited excellent mechanical properties, which can be regarded as one of the best performances among the reported low-k polymer dielectric materials" and "This study proposed polymer dielectric materials and suitable process technologies for the utilization as low-k interconnects, and also provided design rules for ultralow-k materials". We are grateful to the reviewer for the insightful questions and constructive comments, which have inspired us to conduct additional experiments to further strengthen our conclusion. In particular: i) We have conducted additional growth experiments to understand the possible growth mechanism of the 2DP-F films. ii) We developed an electrode transfer method to confirm the dielectric constant and breakdown of the 2DP-F film without the insulating ALD Al₂O₃ layer. iii) we have conducted GIWAXS and time-dependent XPS measurements to confirm the structure of the 2DP-F film. iv) We have systematically conducted more measurements and analyzed the MoS₂ FET devices with and without 2DP-F layer.

We have made our best effort to address all comments from the reviewer, and the updates made in the revised manuscript are highlighted in red.

1. In Fig. 1d, the thickness has linear relationship with time, but it seems that there is a ramp time. For example, the deposited thickness is negligible after 2~3 hours at the growth temperature of 160 and 180 C. Can authors explain them in terms of the deposition mechanism?

Response: We thank the reviewer for raising this interesting question. Motivated by this question, we have performed more growth under different conditions to better understand the growth mechanism.

The possible reaction mechanism for the deposition of oligomers on the substrate from the vapor phase reaction of the monomers and the associated deposition rate is related to the nucleation of the oligomers on the film as well as the reaction rate of the monomers to form oligomers. As evidenced by the ramping growth rate of 2DP-F film when using pre-deposited 2DP-F film as a growth substrate and using 3M AcOH (aq) in a bubbler as catalyst, the oligomers nucleate easier on a pre-deposited 2DP-F film with a thickness of 5 nm, and the thickness can increase to 50 nm after 4h reaction under 160°C, which is three times thicker

than the 2DP-F film deposited on a bare silicon substrate under the same condition.

These results are provided in Figure S4 and discussed in the main text, on page 3

“The growth rate can be increased when using 3M AcOH in a bubbler as the catalyst and using a 5nm- thick 2DP-F film as growth substrate. Under this condition, the thickness of the film can reach more than 500 nm within 8 hours while maintaining a smooth surface (Figure S4).”

Figure S4. (a) The thickness of the 2DP-F film as a function of growth time and temperature when using 5nm 2DP-F thin film as the starting substrate and 3M AcOH (aq) as a catalyst. (b) AFM image of a 530 nm thick 2DP-F film and (c) corresponding line profile.

2. For low-k interconnects, the thickness higher than micrometer is typically recommended. However, the deposition rate of 2DP-F is quite slow. Is there explanation or strategy how the deposition rate can be improved?

Response: We thank the reviewer for this valuable question. To get a thicker film, there are two strategies:

1: It is known that Lewis-acid can catalyze the imine-condensation reaction. We have equipped the system with a bubbler filled with 3M acetic acid solution, and we can get a film of more than 500 nm in thickness within 8 hours.

2: The deposition rate can be increased using a pre-deposited film as a growth substrate. As described in our response to the first comment, the thickness of the film can be dramatically increased to more than 500nm within 8 hours when using pre-deposited 5nm 2DP-F film as substrate and 3M AcOH as catalyst under 200°C growth temperature.

These results are provided in Figure S4 and discussed in the main text on page 3.

“The growth rate can be increased when using 3M AcOH in a bubbler as the catalyst and using a 5nm- thick 2DP-F film as a growth substrate. Under these conditions, the thickness of the film can reach more than 500 nm within 8 hours while maintaining a smooth surface (Figure S4).”

3. The authors claimed that the 2DP-F is an amorphous structure, but there are some clear peaks in XRD data (Fig. S7). What is the origin of the peaks?

Response: We thank the reviewer for the keen observation. There are two main peaks in the structure. The peak at around 24 degrees is assigned to the π - π stacking of the planar

aromatic rings, which have been widely observed in many amorphous 2D polymers (Nature, 602, 91–95 (2022)). Another peak at around 15 degrees may be assigned to the inter-chain distance of the flexible chains (Figure S10 (d) in the revised SI). We also conducted GIWAXS on 2DP-F films, and the result is similar to the powder XRD results. These results are provided in Figure S10 and discussed in the main text on page 4.

“The 2D feature of 2DP-F film was further confirmed by using the grazing-incident wide-angle X-ray scattering (GIWAXS) technique. A diffuse arc in the q_z direction was observed, and this peak at around 1.6 \AA^{-1} corresponded to a layer distance of 0.393 nm , which is larger than typical interlayer spacing in 2D polymers (Figure S10a-c). The enlarged interlayer spacing can be attributed to the flexibility of the sp^3 carbon in HFDA. These results suggested that 2DP-F aligned parallel to the substrate surface. A weaker peak at around 1.0 \AA^{-1} was also observed, which may be attributed to the spacing between flexible chains.”

Figure S10. (a)GIWAXS scattering 2D image and (b) its intensity profile near $Q_r=0$. (c) molecular structure of a building unit in 2DP-F. (d) PXR of 2DP-F powder collected after the reaction.

4. As the dielectric constant showed the impressive values, they should be firmly confirmed. It seems the dielectric constant values were measured by the parallel plate capacitors with Al_2O_3 . I suggest that the authors should measure the dielectric constant values of 2DP-F without Al_2O_3 and confirm that it shows consistent results. The leakage current issue can

be excluded in sufficiently large thickness of 2DP-F.

Response: We thank the reviewer for this valuable suggestion. We acknowledge that the oxide layer could influence the actual dielectric constant of 2DP-F. To address this concern, we employed a “dry transfer” method (Nature, 557, 696–700 (2018)) to fabricate the MIM device without the Al₂O₃ layer. As a result, we observed a slight decrease in the dielectric constant of the film when the oxide layers were removed. The new results are presented in the revised Figure 2. Further discussion is on page 5 in the revised main text.

“To evaluate the capacitance-voltage (C-V) characteristic of 2DP-F thin films, we fabricated parallel plate capacitors on 2DP-F thin films with three different thicknesses, as illustrated in Figure 2a (top). To prevent the short-circuit of the metal-insulator-metal (MIM) devices when using direct metal deposition, we used a “dry-transfer” method to fabricate the MIM structure (Figure S12a-b). The as-fabricated device is illustrated in Figure 2a (bottom). I-V measurement shows that 2DP-F capacitors have low leakage current under an electric field of 1MV/cm, suggesting robust insulation properties of 2DP-F films (Figure S13). Figure 2b shows the C-V characteristic of 2DP-F thin films with different thicknesses at 1MHz frequency. The dielectric constant of the 2DP-F films can be extracted from the following formula:

$$C = \frac{\epsilon_0 \epsilon A}{d}$$

Given the known electrode area A and 2DP-F thin film thickness d, the dielectric constants of 2DP-F thin films are extracted as 1.83±0.10, 1.84±0.16 and 1.84±0.18 for 15, 21, and 33 nm film thicknesses at 1MHz frequency, respectively. Figure S14 shows that for all 2DP-F thin films, the measured capacitances remain nearly constant with the voltage and slightly decrease with the frequency. The dielectric constant-frequency characteristic is shown in Figure 2c. Taking 15 nm 2DP-F films as an example, the dielectric constant will decrease with the frequency, from 1.89±0.08 at 10 kHz to 1.83±0.10 at 1MHz (Figure 2c). We also fabricated a MIM structure on a thicker film (130 nm) using the direct metal deposition method. The 130 nm 2DP-F film exhibited a dielectric constant of 1.92 and a breakdown field of 1.1 MV/cm. (Figure S15a-c). The increased dielectric constant and decreased breakdown field might be attributed to the unreacted amino groups or hydroxyl groups (Figure S8).”.

Figure 2. (a) Schematic illustration of the structure (top) and optical image (bottom) of the parallel plate capacitor. (b) Voltage-dependent capacitance (C - V) and (c) Relative dielectric constant as a function of frequency (C - f) for parallel plate capacitors of 2DP-F thin films with varied thicknesses. (d) schematic illustration of 2DP-F thin films grown in the gaps of interdigitated capacitors. The left figure illustrates the cross-section of interdigitated capacitors in the dashed square. (e) TEM image (top) and EDS elemental mapping of the cross-section of the interdigitated capacitors. (f) Dielectric constant determination from the Silvaco TCAD simulation. The black dashed line corresponds to the dielectric constant of the interdigitated capacitors as a function of the dielectric constant of the gap-filling materials. Red dots correspond to the measured capacitance of the capacitors after filling the gaps with 2DP-F materials. Inset is a modeled equivalent circuit of impedance behavior in the interdigitated capacitors.

5. Please make sure that the I-V characteristics of capacitors in Fig. S10 were obtained with Al₂O₃ or 2DP-F only. Also, Y-axis should be represented by leakage current per unit area (leakage current density) rather than leakage current. What about the out-of-plane breakdown field of the 2DP-F?

Response: We appreciate the reviewer's insightful question and valuable suggestion. We acknowledge that the oxide layer could influence the leakage current measurement of the 2DP-F film. To address this issue, we used a dry transfer method to fabricate the MIM device mentioned in question 5 and measured the leakage current without the Al₂O₃ layer. The results indicate that the leakage current density is below 10^{-7} A/cm² at an electric field of 1MV/cm for both 2DP-F films, and their breakdown electric field is approximately 1.6 MV/cm—higher than the breakdown field of around 0.3 MV/cm in the in-plane direction (Figure S17). This may be due to the increased layer distance introduced by the flexible building blocks in the structure.

These results are provided in Figure S13 and discussed in the main text on page 5.

“To prevent the short-circuit of the metal-insulator-metal (MIM) devices when using direct metal deposition, we used a “dry-transfer” method to fabricate the MIM structure (Figure S12a-b).³⁷ The as-fabricated device is illustrated in Figure 2a (bottom). I-V measurement shows that 2DP-F capacitors have low leakage current for under an electric field of 1MV/cm, suggesting robust insulation properties of 2DP-F films (Figure S13).”

Figure S13. Leakage current density versus the applied electrical field across the MIM devices based on 2DP-F with different thicknesses.

6. The dielectric constant was measured to be slightly, but consistently increased according to the thickness of the 2DP-F. The explanation should be provided.

Response: We thank the reviewer for this valuable comment. After carefully measuring the dielectric constant of 2DP-F films of different thicknesses without the Al₂O₃ layer, we found that their dielectric constants are very similar. Although there is a slight increase in the dielectric constant, this may be attributed to unreacted amino or hydroxyl groups. Thickness-dependent XPS analysis (Figure S8) revealed the presence of some unreacted amino groups, which could contribute to the increase in the dielectric constant due to their polar nature. These results are provided in Figure S8 and discussed in the main text, on page 5.

“The increased dielectric constant and decreased breakdown field might be attributed to the unreacted amino groups or hydroxyl groups (Figure S8).”

Figure S8. Thickness-dependent XPS N1s spectra of 2DP-F films. The result indicates that some unreacted amino groups are present when the thickness is larger than 100 nm.

7. The authors claimed that the excellent mechanical properties can be attributed to the covalently crosslinked structure and 2-dimensional structure of 2DP-F. Can crosslinking density be calculated through chemical analysis? Also, according to the authors' explanation and analysis, I understood that 2DP-F has a horizontal (face-on) orientation in two dimensions. Could you analyze the orientation of 2DP-F or provide a suitable scheme and explanation?

Response: We thank the reviewer for this valuable comment. We conducted an XPS N1s spectrum analysis to confirm the crosslink density of the 2DP-F film. The results showed that when the film thickness is below 60 nm, no significant -NH_2 peak is observed, indicating a very high crosslink density. As the film thickness increases to above 120 nm, the -NH_2 peak becomes apparent. For a 120 nm film, the ratio of the -NH_2 peak to the amide peak is approximately 1:200, indicating a crosslink density greater than 99.5%. In thicker films, the ratio is around 1:120, corresponding to a crosslink density greater than 99.2%. These results are provided in Figure S8 and discussed in the main text on page 4.

“It is worth noting that the crosslink density of 2DP-F decreases as the film thickness increases. Some defects corresponding to unreacted amino groups can be observed in the N1s XPS spectrum. In a 500 nm film, the ratio of unreacted amino groups to amide groups is approximately 1:120, indicating a crosslink density of over 99.5% (Figure S8).”

Figure S8. Thickness-dependent XPS N1s spectra of 2DP-F films. The result indicates that some unreacted amino groups are present when the thickness is larger than 100 nm.

To confirm the orientation of 2DP-F, we conducted GIWAXS to analyze the structure of 2DP-F films, as described in comment 3. The GIWAXS patterns indicate the face-on orientation of 2DP-F films, and these results are provided in Figure S10 and discussed in the main text on page 4.

“The 2D feature of the 2DP-F film was further confirmed by using the grazing-incident wide-angle X-ray scattering (GIWAXS) technique. A diffuse arc in the q_z direction was observed, and this peak at around 1.6 \AA^{-1} corresponded to a layer distance of 0.393 nm , which is larger than typical interlayer spacing in 2D polymers (Figure S10a-c). The enlarged interlayer spacing can be attributed to the flexibility of the sp^3 carbon in HFDA. These results suggested that 2DP-F aligned parallel to the substrate surface. A weaker peak at around 1.0 \AA^{-1} was also observed, which may be attributed to the spacing between flexible chains.”

Figure S10. (a)GIWAXS scattering 2D image and (b) its intensity profile near $Q_r=0$. (c) molecular structure of a building unit in 2DP-F. (d) PXRD of 2DP-F powder collected after the reaction.

8. The 2DP-F exhibited ultra-low-k as well as superior mechanical properties compared to polymers and porous OSGs (Fig. 3g). It has been reported that 2D hexagonal boron nitride (hBN) has low dielectric constant and excellent mechanical properties. Can the authors describe the advantages of 2DP-F compared to 2D hBN?

Response: We thank the reviewer for the valuable comment. Compared to h-BN, there are several advantages of 2DP-F as described below:

1. Growth conditions: The dielectric constant of crystalline h-BN is around 3.9, and for amorphous h-BN, it is approximately 1.4. However, most reported CVD-grown h-BN requires very high growth temperatures ($>400^\circ\text{C}$) and high vacuum conditions, which are not compatible with current microelectronic fabrication processes. In contrast, 2DP-F films can be grown at relatively low temperatures, even below 160°C , under ambient conditions.

2. Low-k Dielectric Application: One of the goals of using low-k dielectrics is to reduce interconnect crosstalk and RC delay. Therefore, direct deposition and complete filling of low-k dielectric materials into interconnects is crucial. 2DP-F can be conformally grown onto arbitrary substrates and fill small gaps within interconnects, which is challenging for h-BN.

3. Patterning and Versatility: As a low-temperature-grown polymeric material, 2DP-F

films can be easily patterned into desired structures using a patterned substrate or oxygen plasma (Figure S6).

4. Design Flexibility: We employed a bottom-up method to prepare 2DP-F, allowing for a highly customizable structures compared to h-BN. The dielectric constant of the films can be tuned by using different monomers, and the structure-dielectric relationship of 2D polymers can be further explored.

I) Direct photolithography

II) Deposition on pre-patterned substrates

Figure S6. (a) Schematic illustration of direct photolithography on 2DP-F films. (b) Optical image of patterned 2DP-F films. (c) AFM image of 2DP-F film before and after patterning and (d) Corresponding line profile. (e) Schematic illustration of patterning 2DP-F by direct deposition of 2DP-F on patterned substrates. (f) Optical image of 2DP-F film after lift-off in acetone with ultra sonification. (g) AFM image of the area highlighted in black square and (h) Corresponding line profile.

9. It should be better that the trap density of 2D MoS₂ FETs with 2DP-F is quantitatively analyzed compared to that without 2DP-F.

Response: We thank the reviewer for this valuable suggestion. In response, we have

calculated the trap density of the MoS₂ FETs based on their subthreshold swing. The results are now included on page 9 of the revised manuscript.

“To further probe the interface properties, the trap density (D_{it}) between MoS₂/2DP-F and MoS₂/HfO₂ was estimated using the following expression from the subthreshold swing (SS):

$$SS = \frac{\ln(10) k_b T}{q} \left(1 + \frac{q D_{it}}{C_g}\right)$$

Where k_b is the Boltzman constant, q is the elementary charge and C_g is the gate capacitance. The extracted D_{it} is around $6.2 \times 10^{13} \text{ cm}^{-2} \cdot \text{eV}^{-1}$ and $1.1 \times 10^{14} \text{ cm}^{-2} \cdot \text{eV}^{-1}$ for MoS₂/2DP-F and MoS₂/HfO₂ interface, respectively.”

10. Compared to those without 2DP-F, the mobility variation of 2D MoS₂ FETs with 2DP-F is considerable. Based on the results, it seems that there are potential issues with the process uniformity or reproducibility of 2DP-F.

Response: We thank the reviewer for the insightful comments. In our original manuscript, we were using CVD grown MoS₂ crystals from different batches, so the mobility of the FETs was scattered. In the revised manuscript, we fabricated our devices based on monolayer MoS₂ crystals from the same batch, and the new results are provided in revised Figure 4. In our new results, we measured 15 new devices with and without a 2DP-F layer using the same batch of CVD-grown MoS₂, and the mobility and on/off ratio data are less scattered.

Figure 4. (a) Optical image of the FET device fabricated on transferred monolayer MoS₂/2DP-F thin film heterostructure. (b) The I_d - V_{ds} characteristic of the FET devices. (c) Transfer characteristics of the devices with and without 2DP-F dielectric layer. Inset: log-scale transfer characteristic of the devices with and without 2DP-F. (d) The distribution of the mobility and on-off ratio of the FET devices with and without the 2DP-F dielectric layer.

Reviewer #2 (Remarks to the Author):

Summary:

Fang and co-authors reported on the development of a vacuum-deposited two-dimensional polymer (2DP) material, which exhibit low-k dielectric value of approximately 2. Using double-zoned tube furnace and reacting 4,4'-(Hexafluoroisopropylidene)dianiline (HFDA) with triformylphloroglucinol (Tp)/substrate, the authors achieved growth of low-k 2DP film on arbitrary substrate via chemical vapor deposition (CVD). The authors demonstrated that the transistors employing the proposed dielectric exhibit superior electrical performances (i.e. on-current level, carrier mobility, and etc) as compared to those built without their dielectric. However, despite the novel findings of vacuum-deposited 2DP film, the current version of manuscript is not appropriate for publication in Nature Communications unless the authors provide better demonstration of how their method and material is beneficial in terms of device fabrication, as stated in their introductory section. Specific technical comments are as follows:

Response: We thank the reviewer for making thoughtful suggestions and critical comments. We have made our best effort to address all these comments, and all updates made in the revised manuscript are highlighted in red.

Technical Comments:

Comment #1: The authors mention in the introduction that for integrated circuits (ICs), the development of low-k dielectric with robust mechanical properties are desired to reduce the RC delay. Yet in their manuscript, there is no evidence that their developed low-k dielectric material is capable of achieving such reduced RC delay in ICs. Please provide proper evidence that the proposed material can be used to achieve such breakthrough. It is hard to find proper correlation between what the authors are claiming and what the authors are actually presenting.

Response: We thank the reviewer for this critical comment. The RC (resistor-capacitor) time constant $\tau = R \times C$ represents the time required to charge a capacitor from 0 voltage to 63.2% of the value of an applied DC voltage V_0 , where $V(t) = V_0(1 - e^{-t/\tau})$. As feature sizes decrease in microelectronic circuits, RC delay becomes a significant factor because interconnect capacitance increases dramatically.

To address this issue, the RC delay can be reduced by 1). Lowering the resistance of the interconnect, for example, by replacing aluminum wires with more conductive copper wires, or 2) decreasing the interconnect capacitance by replacing typical silicon dioxide with low dielectric constant materials. Thus, the interconnect capacitance can be reduced.

To evaluate the impact of using 2DP-F as an interconnect dielectric, a Silvaco TCAD simulation was carried out. The simulated structure is shown in Figure S18. The results indicate that replacing silicon dioxide ($\kappa \sim 4.0$) with 2DP-F ($\kappa \sim 1.8$) reduces the interconnect capacitance by 52.8%. Additionally, when compared to SILK, a low-k polymer with a dielectric constant around 2.6, 2DP-F reduces the interconnect capacitance by 28.9%. In the

simulated structure, with the resistance of the aluminum wire around 44 ohms, the RC delay constant is reduced from 8.67×10^{-16} s and 5.76×10^{-16} s to 4.09×10^{-16} s when replacing the interconnect silicon oxide and SILK with 2DP-F, respectively. Thus, the RC time constant can be reduced by 52.8% and 28.9% when replacing SiO_2 and SILK with 2DP-F as interconnect dielectric, respectively.

These results are provided in Figure S18 and discussed in the main text, on page 6.

“To evaluate the impact of using 2DP-F as ICD to reduce the RC delay constant, a Silvaco TCAD simulation is carried out. The simulated structure is shown in Figure S18a. In this simulation, the length of the metal wires is 100nm, and the width and height of the metal wires are fixed at 8nm. The total parasitic capacitance in the middle metal wire is simulated when applying different ICDs with different dielectric constants.

As demonstrated in Figure S18, the total capacitance is reduced by 52.8% and 28.9% when replacing silicon oxide ($k \sim 4.0$) and SILK[®] (trademark of The Dow Chemical Company, $k \sim 2.6$) with 2DP-F ($k \sim 1.8$) as the ICD, respectively, as the device feature scales down (S18b). The resistance of the simulated aluminum wire is 44Ω , resulting in a reduction of the RC delay constant ($\tau = R \times C$) from 8.67×10^{-16} s and 5.76×10^{-16} s to 4.09×10^{-16} s when replacing the interconnect silicon oxide and SILK[®] with 2DP-F, respectively. These simulation results highlight the potential of 2DP-F as an interconnect dielectric to reduce the RC delay constant as the features of the devices scale down.”

Figure S18. (a). Schematic of a simplified Cu interconnect structure (top) and corresponding equivalent circuit used for the Silvaco TCAD simulation. (b) Interconnect parasitic capacitance reduction by substituting silicon oxide and SILK with 2DP-F.

Comment #2: One of the main reasons for developing vacuum-based deposition method for 2DPs is believed to be associated with compatibility to current microfabrication techniques for device fabrication, to grant large-area uniformity, sequential and in-situ deposition of other functional layers, followed by their processing via techniques such as photolithography, etching, and etc. For this, the developed dielectric requires high stability in terms of chemical,

thermal, and mechanical performances. These aspects are missing in the manuscript so please provide relevant data.

Response: We thank the reviewer for the comment. We agree that the 2DP-F films should be chemically, thermally, and mechanically stable during the microfabrication process. To verify 2DP-F films' stability during a typical microfabrication process, we performed a standard photolithography process on a 20nm 2DP-F film. Our results indicate that the 2DP-F film can be effectively patterned, and its surface morphology remains unchanged after the process. Additionally, we grew 2DP-F directly on a pre-patterned substrate and then lifted off the film in acetone with ultrasonication. The remaining pattern exhibited a very smooth surface and well-defined edges. These results demonstrate that 2DP-F is compatible with and highly stable under current typical microfabrication processes.

These results are provided in Figure S6, and further discussion is highlighted in the revised main text on page 3.

"The CVD grown 2DP-F exhibited good thermal, mechanical and chemical stability, and they can be patterned by a standard photolithography or deposited on pre-patterned substrates (Figure S6)."

Figure S6. (a) Schematical illustration of direct photolithography on 2DP-F films. (b) Optical image of patterned 2DP-F films. (c) AFM image of 2DP-F film before and after patterning and (d) Corresponding line profile. (e) Schematic illustration of patterning 2DP-F by direct deposition of 2DP-F on patterned substrates. (f) Optical image of the 2DP-F film after lift-off in acetone with ultrasonication. (g) AFM image of the area highlighted in black square and (h) corresponding line profile.

Comment #3: In addition to Comment #2, the authors need to provide statistical data in terms of device uniformity (transistor performance) when produced in large-scale. It appears in Figure 4d that there is a substantial variation in the carrier mobility of transistors fabricated using the proposed dielectric, and the authors need to specify whether it is attributed to the dielectric performance or other causes are involved.

Response: We thank the reviewer for the insightful comment. In the original manuscript, we used CVD-grown MoS₂ from different batches, which led to scattered mobility results. In the

revised manuscript, we fabricated our devices using monolayer MoS₂ crystals from the same batch. The updated results are shown in Figure 4 of the revised manuscript. We measured 15 new devices with and without the 2DP-F layer using MoS₂ from the same batch, resulting in less scattered data for mobility and on/off ratio.

Figure 4. (a) Optical image of the FET device fabricated on transferred monolayer MoS₂/2DP-F thin film heterostructure. (b) The I_d - V_{ds} characteristic of the FET devices. (c) Transfer characteristics of the devices with and without 2DP-F dielectric layer. Inset: log-scale transfer characteristic of the devices with and without 2DP-F. (d) The distribution of the mobility and on-off ratio of the FET devices with and without the 2DP-F dielectric layer.

Comment #4: Once again, as briefly commented in Comment #2, one of the main reasons for developing vacuum-deposited dielectric layer should be compatibility with current microfabrication techniques. It seems from the structures and fabrication methods for the parallel plate capacitors and transistors that when metal electrodes are directly deposited onto the proposed dielectric layer, it may be severely damaged. Please clarify this issue.

Response: We thank the reviewer for the comment. We acknowledge that e-beam deposition can damage our 2DP-F thin films, which is a common issue with many organic thin films and ultrathin dielectrics. To address this, we have implemented two solutions:

1. We prepared a relatively thick film (130nm) and directly deposited electrodes on it. For thicker samples, the dielectric film maintained its low dielectric constant and insulating property after metal deposition (Figure S15).

Figure S15. (a) Capacitance-voltage (C-V) characteristic, (b) C-f characteristic and (c) leakage current of 2DP-F MIM device prepared using the direct metal deposition method.

2. For relatively thin films (<50 nm), we used a dry transfer method to transfer prefabricated electrodes on them. This method was widely used to prepare high-performance 2D materials devices (Nature, 557, 696–700 (2018)). We measured the dielectric constant and breakdown of the 2DP-F films using this method. The new results are provided in revised figure 2 and figure S12 and S13. Further discussion is included in the main text on page 5.

“To evaluate the capacitance-voltage (C-V) characteristic of 2DP-F thin films, we fabricated parallel plate capacitors on 2DP-F thin films with three different thicknesses, as illustrated in Figure 2a (top). To prevent the short-circuit of the metal-insulator-metal (MIM) devices when using direct metal deposition, we used a “dry-transfer” method to fabricate the MIM structure (Figure S12a-b).³⁷ The as-fabricated device is illustrated in Figure 2a (bottom). I-V measurement shows that 2DP-F capacitors have low leakage current for under an electric field of 1MV/cm, suggesting robust insulation properties of 2DP-F films (Figure S13). Figure 2b shows the C-V characteristic of 2DP-F thin films with different thicknesses at 1MHz frequency. The dielectric constant of the 2DP-F films can be extracted from the following formula:

$$C = \frac{\epsilon_0 \epsilon A}{d}$$

Given the known electrode area A and 2DP-F thin film thickness d, the dielectric constants of 2DP-F thin films are extracted as 1.83 ± 0.10 , 1.84 ± 0.16 and 1.84 ± 0.18 for 15, 21 and 33 nm film thicknesses at 1MHz frequency, respectively. Figure S14 shows that for all 2DP-F thin films, the measured capacitances remain nearly constant with the voltage and slightly decrease with the frequency. The dielectric constant-frequency characteristic is shown in Figure 2c. Taking 15 nm 2DP-F films as an example, the dielectric constant will decrease with the frequency, from 1.89 ± 0.08 at 10 kHz to 1.83 ± 0.10 at 1MHz (Figure 2c). We also fabricated a MIM structure on a thicker film (130 nm) using the direct metal deposition method. The 130 nm 2DP-F film exhibited a dielectric constant of 1.92 and a breakdown field of 1.1 MV/cm. (Figure S15a-c). The increased dielectric constant and decreased breakdown field might be attributed to the unreacted amino groups or hydroxyl groups (Figure S8).”.

Figure 2. (a) Schematic illustration of the structure (top) and optical image (bottom) of the parallel plate capacitor. (b) Voltage-dependent capacitance (C-V) and (c) Relative dielectric constant as a function of frequency (C-f) for parallel plate capacitors of 2DP-F thin films with varied thicknesses. (d) schematic illustration of 2DP-F thin films grown in the gaps of interdigitated capacitors. The left figure illustrates the cross-section of interdigitated capacitors in the dashed squares. (e) TEM image (top) and EDS elemental mapping of the cross-section of the interdigitated capacitors. (f) Dielectric constant determination from the Silvaco TCAD simulation. The black dashed line corresponds to the dielectric constant of the interdigitated capacitors as a function of the dielectric constant of the gap-filling materials. Red dots correspond to the measured capacitance of the capacitors after filling the gaps with 2DP-F materials. Inset is a modeled equivalent circuit of impedance behavior in the interdigitated capacitors.

i) Spin coat PMMA; ii) thermal release tape; iii) Peel off electrode in the presence of 10% NaOH(aq); iv) Put TRT/PMMA/Electrode on 2DP-F film; v) Remove TRT and PMMA by heating to 120°C and acetone respectively.

Figure S12. Schematical illustration of the "dry-transfer" method to fabricate MIM devices based on 2DP-F.

Figure S13. Leakage current density versus the applied electrical field across the MIM devices based on 2DP-F with different thicknesses.

Reviewer #3 (Remarks to the Author):

Fang et al report a chemical-vapor-deposition method for growing two-dimensional polymer on arbitrary substrates. While the properties of the polymers are consistently well-regulated, the manuscript still leaves some room for doubt.

Response: We thank the reviewer for making thoughtful suggestions and critical comments. We have made our best effort to address all these comments, and all updates made in the revised manuscript are highlighted in red.

1. In the manuscript, the author defines 2DP-F as a two-dimensional polymer. Whether the polymer has two-dimensional molecular structure, not a cross-linked polymer, can it be proven with experimental data? In addition, it's necessary to supplement the pore size and porosity of 2DP-F for analyzing whether there are so many micropores contributing to the low dielectric constant performance of the polymer.

Response: We thank the reviewer for the insightful question. We acknowledge that proving 2DP-F is a 2D polymer rather than a crosslinked polymer is challenging. To address this, we performed several experiments to determine its structure:

1): GIWAXS Analysis: We conducted Grazing-Incidence Wide-Angle X-ray Scattering (GIWAXS) characterization on the 2DP-F film, which revealed a diffuse arc along the q_z axis, confirming an interlayer spacing of approximately 0.393 nm in the z-direction. This pattern is characteristic of the face-on orientation of 2D polymers (Nature, 602, 91–95 (2022)). These results are provided in Figure S10 and discussed in the main text on page 4.

“The 2D feature of 2DP-F film was further confirmed by using the grazing-incident wide-angle X-ray scattering (GIWAXS) technique. A diffuse arc in the q_z direction was observed, and this peak at around 1.6 \AA^{-1} corresponded to a layer distance of 0.393 nm, which is larger than typical interlayer spacing in 2D polymers (Figure S10a-c). The enlarged interlayer spacing can be attributed to the flexibility of the sp^3 carbon in HFDA. These results suggested that 2DP-F aligned parallel to the substrate surface. A weaker peak at around 1.0 \AA^{-1} was also observed, which may be attributed to the spacing between flexible chains.”

Figure S10. (a)GIWAXS scattering 2D image and (b) its intensity profile near $Q_r=0$. (c) molecular structure of a building unit in 2DP-F. (d) PXRD of 2DP-F powder collected after the reaction.

2): **Swelling Test:** We performed a swelling test on a relatively thick 2DP-F film (around 150 nm), a common method for determining crosslink density. The film's thickness was measured before and after immersion in acetonitrile overnight. The thickness and surface morphology did not change as indicated by AFM, distinguishing it from conventional crosslinked polymers (Macromolecules 2023, 56, 11, 4000–4011).

Figure R1. The AFM image and corresponding line profile of a 150 nm 2DP-F film before and after swelling test.

3): **Leakage Current Analysis:** Our leakage current measurements showed significant different breakdown fields for 2DP-F in the in-plane (0.3 MV/cm) and out-of-plane (1.5 MV/cm) directions, which may be attributed to the face-on configuration observed in the GIWAXS results.

Figure S13. Leakage current density versus the applied electrical field across the MIM devices based on 2DP-F with different thicknesses.

Figure S17. (a). C-V characteristic of interdigitated capacitors before and after gap filling. (b) breakdown of the interdigitated capacitors.

We also conduct several experiments to confirm the porosity of the 2DP-F film. Direct surface area measurement of thin films is challenging. Instead, we performed BET analysis on powder collected from the reactor wall after the reaction. The 2DP-F powder exhibited a surface area of 958 cm²/g and a pore volume of 0.57 cc/g, indicating high porosity. These results are provided in Figure S11 and discussed in the revised main text, on page 5.

“The porosity nature of 2DP-F was confirmed by Brunauer-Emmett-Teller (BET) analysis. The surface area of 2DP-F powder reached 958 m²/g and a pore volume of 0.57cc/g, indicating its high porosity (Figure S11).”

Figure S11. Nitrogen sorption curves for 2DP-F powder.

2. The dielectric properties of 2DP-F are affected by the presence or absence of unreacted hydroxyl and amino groups.

Response: We thank the reviewer for the insightful comment. In response, we investigated the layer-dependent structure of 2DP-F by performing XPS analysis on films of varying thicknesses. The results showed that for 2DP-F films with a thickness below 60 nm, there were no significant unreacted amino groups. As the thickness increased, unreacted amino groups became detectable, with the ratio of unreacted amino groups to amide groups rising to approximately 1:120 in a 500 nm thick 2DP-F film. For films thinner than 50 nm, there was no significant change in the dielectric constant. However, when the thickness increased to 130 nm, the dielectric constant rose to 1.92, which may be attributed to the presence of unreacted groups. These results are provided in Figure S8 and discussed in the main text on page 4 and page 5.

“It is worth noting that the crosslink density of 2DP-F decreases as the film thickness increases. Some defects corresponding to unreacted amino groups can be observed in the N1s XPS spectrum. In a 500 nm film, the ratio of unreacted amino groups to amide groups is approximately 1:120, indicating a crosslink density of over 99.5% (Figure S8).”

“We also fabricated a MIM structure on a thicker film (130 nm) using the direct metal deposition method. The 130 nm 2DP-F film exhibited a dielectric constant of 1.92 and a breakdown field of 1.1 MV/cm (Figure S15a-c). The increased dielectric constant and decreased breakdown field might be attributed to the unreacted amino groups or hydroxyl groups (Figure S8).”

Figure S8. Thickness-dependent XPS N1s spectra of 2DP-F films. The result indicates that some unreacted amino groups are present when the thickness is larger than 100 nm.

3. In the chapter analyzing the mechanical properties of the 2DP-F film, as the thickness of

the polymer film increases, the mechanical performance decreases. The author explains this is due to weak interlayer interactions. However, it is also possible that 2DP-F is a cross-linked polymer, and as the thickness increases, the defects increase, leading to the degradation of the entire film's mechanical strength.

Response: We thank the reviewer for the insightful question. As discussed in our response to comment 1, we used GIWAXS on 2DP-F film to confirm its 2D features. However, as mentioned in our response to comment 2, the defect density will increase as the 2DP-F film thickness increases. Thus, the reduction in mechanical properties may be due to both the weak interlayer interactions observed in 2D materials and the increased defect density. We have clarified this possibility in the revised manuscript.

"The decrease in mechanical properties is possibly attributed to the weak interlayer interactions observed in many 2D materials and the increased density of defects as the film thickness increases (Figure S8)"

4. In the introduction, the author emphasizes the potential of the high thermal conductivity of 2D polymers in high-density interconnect applications. May I know what are the in-plane and through-plane thermal conductivities of 2DP-F? Are they higher than the reported low-dielectric constant polymers?

Response: We thank the reviewer for the valuable comment. To measure the thermal conductance of 2DP-F, we conducted a Time-Domain Thermoreflectance (TDTR) experiment on a 130 nm thick film. The results indicated a high out-of-plane thermal conductance of approximately 0.38 W/m·K, which is comparable to most MOF materials and slightly lower than crystalline COFs (~1 W/m·K). The lower out-of-plane thermal conductance compared to crystalline COF might be attributed to the increased interlayer distance with the introduction of flexible HFDA building blocks in 2DP-F. This value is higher than that of most low-k polymers. The in-plane thermal conductance of the 2DP-F film remains to be explored further due to technical limitations.

These results are provided in Figure S21 and discussed in the main text, page 9.

"In addition to its excellent mechanical properties, 2DP-F also exhibits good out-of-plane thermal conductance of 0.38 W/m·K (Figure S21). This value is comparable to many MOF materials and higher than most low-k polymers. However, due to the increased interlayer distance with the introduction of the flexible HFDA building blocks in 2DP-F, this value is lower than that of crystalline 2D COFs."

Figure S21. (a) Phase lag vs. frequency data obtained from FDTR measurements shows a good approximation to the calculated best-fit curve. Each measurement is an average of three runs. (inset: schematic of FDTR measurement). (b) Sensitivity analysis of the thermal conductivity k of the COF, the thermal boundary conductance $G1$ between Au and the COF, and $G2$ between the COF and the substrate. k of the COF is highly sensitive throughout the frequency range of our measurement.

5. In Figure S10, it can't be ruled out that the low leakage current density is contributed by the layer of aluminum oxide, rather than the body of the polymer film.

Response: We appreciate the reviewer's insightful question and valuable suggestion. We acknowledge that the oxide layer could influence the leakage current measurement of the 2DP-F film. To address this issue, we used a dry transfer method to fabricate the MIM device without the Al_2O_3 layer and measured the leakage current. The results indicate that the leakage current density is below 10^{-7} A/cm² at an electric field of 1MV/cm for both 2DP-F films, and their breakdown electric field is approximately 1.6 MV/cm—higher than the breakdown field of around 0.3 MV/cm in the in-plane direction (Figure S17). This may be due to the increased layer distance introduced by the flexible building blocks in the structure. These results are provided in Figure S13 and discussed in the main text on page 5.

"To prevent the short-circuit of the metal-insulator-metal (MIM) devices when using direct metal deposition, we used a "dry-transfer" method to fabricate the MIM structure (Figure S12a-b).³⁷ The as-fabricated device is illustrated in Figure 2a (bottom). I-V measurement shows that 2DP-F capacitors have low leakage current for under an electric field of 1MV/cm, suggesting robust insulation properties of 2DP-F films (Figure S13)."

Figure S13. Leakage current density versus the applied electrical field across the MIM devices based on 2DP-F with different thicknesses.

6. In this sentence "which is confirmed by the bright-field TEM image and EDS elemental mapping (Figure 3e) ", the Figure 3e does not correspond to the related description.

Response: We thank the reviewer for pointing this out. The bright-field TEM image and EDS elemental mapping should be in Figure 2e. We have corrected this mistake accordingly in the revised manuscript.

REVIEWER COMMENTS

Reviewer #1 (Remarks to the Author):

I acknowledge the authors have addressed the questions and comments that I raised, by providing additional experimental evidence and explanations. The results from the original manuscript, along with the additional data provided during the revision process, strongly support that the 2DP-F polymer developed in this study exhibits a low dielectric constant in addition to excellent electrical and mechanical properties. Furthermore, the authors have demonstrated its practical applicability by applying the 2DP-F polymer to 2D MoS₂ transistors. Therefore, I now agree to publish this revised manuscript in Nature Communications.

Response: We highly appreciate the reviewer's efforts in evaluating our work, which we believe has greatly improved the quality of our manuscript. Thank you for your positive recommendation.

I was also asked to comment on the authors' revisions in response to the concerns raised by referee #3 during the previous round of review.

Response: Thank you for your effort.

In the 1st round of revision for Fang et al., the Reviewer #3 evaluated that the manuscript is well-regulated, but suggested some comments to improve the manuscript, as follows.

In the 1st comment, the Reviewer suggested to provide experimental evidence of two-dimensional structure of the 2DP-F polymer and to supplement the porosity of 2DP-F with its pore size. The authors provided the GIWAXS data to prove its two-dimensional structure with the proper reference and provided BET analysis data to present its high porosity with the exact value of pore volume.

Response: Thank you.

In the 2nd comment, the Reviewer mentioned that the dielectric properties of 2DP-F can be affected by unreacted hydroxyl and amino groups. The authors provided the experimental data that confirmed the amount of unreacted amino groups increased with the increasing thickness of the 2DP-F polymer. Nevertheless, most of amino groups reacted in the thickest film (~500 nm), and the dielectric constant was still very low (~1.92) with the increased thickness of 2DP-F (~130 nm).

Response: Thank you.

In the 3rd comment, the Reviewer suggested that the decreased mechanical performance of the 2DP-F polymer with increased thickness may result from increased defects. The authors acknowledged this comment and included this possibility in the revised manuscript.

Response: Thank you.

In the 4th comment, the Reviewer required the thermal conductivity of the 2DP-F. The authors analyzed thermal conductivity by TDTR experiment, and provided out-of-plane thermal

conductance of the 2DP-F, which can be regarded as a higher value than most low-k polymers. One concern is that the references on thermal conductivity values for low-k polymers were outdated. I suggest that the authors find recent studies on low-k polymers and compare the thermal conductivity value. Also, the Reviewer was curious about in-plane and through-plane thermal conductivity values. I recommend that the authors provide the detailed explanation regarding technical limitations.

Response: Thank you for your comment and suggestion. In response, we have added several new references related to the thermal conductivity of metal-organic frameworks (MOFs) and polymers (references 58-65). Regarding to the in-plane thermal conductivity measurements, we are actively collaborating with Prof. Zhiting Tian's group at Cornell University to directly measure this property in other COF materials. However, since this work is ongoing and has not yet been published, we have chosen not to include these results in the current manuscript to preserve the novelty of our ongoing study. We appreciate the reviewer's understanding on this matter.

In the 5th comment, the Reviewer mentioned that the leakage current density can be affected by aluminum oxide layer. The authors made a reasonable response by providing leakage current property of the MIM device with 2DP-F polymer.

Response: Thank you.

In the last comment, the Reviewer pointed out the typo on the figure number, and the authors revised it.

Response: Thank you.

Reviewer #2 (Remarks to the Author):

The authors have well prepared their revised manuscript, and it appears that the concerns raised by the reviewer has been clarified with empirical and simulation data. Although it is still a bit ambiguous as to how the developed method and material is beneficial in terms of device fabrication, the design rules for ultralow-k dielectric are valid and maybe used to explore ICs with low-k interconnects. In this regard, the reviewer feels that the manuscript deserves publication in Nature Communications without further revisions.

Response: We highly appreciate the reviewer's efforts in evaluating our work, which we believe has greatly improved the quality of our manuscript. Thank you for your positive recommendation.